# Study on the Properties of Partially Transparent Wood under Different Delignification Processes

**DOI:** 10.3390/polym12030661

**Published:** 2020-03-15

**Authors:** Yan Wu, Jichun Zhou, Qiongtao Huang, Feng Yang, Yajing Wang, Jing Wang

**Affiliations:** 1College of Furnishings and Industrial Design, Nanjing Forestry University, Nanjing 210037, China; 15250988513@163.com (J.Z.); lionzaka@163.com (Y.W.); wangjing_9711@163.com (J.W.); 2Co-Innovation Center of Efficient Processing and Utilization of Forest Resources, Nanjing Forestry University, Nanjing 210037, China; 3Department of Research and Development Center, Yihua Lifestyle Technology Co., Ltd., Shantou 515834, China; huangqt@yihua.com; 4Fashion Accessory Art and Engineering College, Beijing Institute of Fashion Technology, Beijing 100029, China

**Keywords:** transparent wood, orthogonal test, partial delignification, light transmittance

## Abstract

Two common tree species of Betula alnoides (*Betula*) and New Zealand pine (*Pinups radiata D. Don*) were selected as the raw materials to prepare for the partially transparent wood (TW) in this study. Although the sample is transparent in a broad sense, it has color and pattern, so it is not absolutely colorless and transparent, and is therefore called partially transparent. For ease of interpretation, the following “partially transparent wood” is referred to as “transparent wood (TW)”. The wood template (FW) was prepared by removing part of the lignin with the acid delignification method, and then the transparent wood was obtained by impregnating the wood template with a refractive-index-matched resin. The goal of this study is to achieve transparency of the wood (the light transmittance of the prepared transparent wood should be improved as much as possible) by exploring the partial delignification process of different tree species on the basis of retaining the aesthetics, texture and mechanical strength of the original wood. Therefore, in the process of removing partial lignin by the acid delignification method, the orthogonal test method was used to explore the better process conditions for the preparation of transparent wood. The tests of color difference, light transmittance, porosity, microstructure, chemical groups, mechanical strength were carried out on the wood templates and transparent wood under different experimental conditions. In addition, through the three major elements (lignin, cellulose, hemicellulose) test and orthogonal range analysis method, the influence of each process factor on the lignin removal of each tree species was obtained. It was finally obtained that the two tree species acquired the highest light transmittance at the experimental level 9 (process parameters: NaClO_2_ concentration 1 wt%, 90 °C, 1.5 h); and the transparent wood retained most of the color and texture of the original wood under partial delignification up to 4.84–11.07%, while the mechanical strength with 57.76% improved and light transmittance with 14.14% higher than these properties of the original wood at most. In addition, the wood template and resin have a good synergy effect from multifaceted analysis, which showed that this kind of transparent wood has the potential to become the functional decorative material.

## 1. Introduction

Transparent wood is usually prepared by impregnating delignified wood template with a refractive-index-matched resin [1]. In recent years, transparent wood, an emerging achievement in wood modification, has attracted attention and research due to its many advantages such as light weight, light transmission, environmental protection, and high mechanical properties. Most of the experiments focus on electronic equipment, optical devices and energy-saving buildings, etc. For example, Li et al. [2] used 2–5 cm thick translucent wood composite material (light transmittance 40%) as the wall material of the house model. The wall material is made by the method of H_2_O_2_ vapor delignification and epoxy resin impregnation, which can effectively capture the external environment light. In addition, unlike the transparent wood roof, the light intensity inside the translucent wood house model is more conducive to people’s daily life. In 2019, Wang et al. [3] used photochromic materials to infiltrate modified wood templates to obtain photochromic transparent wood. This kind of transparent wood appears a bright purple to colorless color change under light and shows about 65% good optical transmittance and 90% high optical haze, which is very important for the application of Smart windows and anti-counterfeiting materials. In the same year, Li et al. [4] successfully assembled the perovskite solar cells treated at low temperature (<150 °C) directly on the transparent wood substrate for the first time, with the power conversion efficiency up to 16.8%, which confirmed that the transparent wood is suitable as a substrate for solar cell modules and has potential in energy-efficient building applications. Celine Montanari et al. [5] also put forward the view that transparent wood should be endowed with multiple functions. Using polyethylene glycol as the base material, PCMs (phase-change materials) with stable shape was embedded in delignified wood templates to obtain functional transparent wood. The application of the heat storage and reversible light transmittance of the transparent wood in thermal energy storage was discussed.

In fact, the development and application of transparent wood reflects the broad development prospects of wood composite materials [6], and the numerical improvement of light transmittance is no longer a single pursuit. On the contrary, people are paying more and more attention to the functionality of transparent wood and its corresponding application fields. At present, most of the transparent or translucent wood tends to be glass, colorless and no texture, and lacks the visual and tactile natural characteristics of wood [7]. Based on the inspiration given by this cutting-edge subject, from another perspective, if wood still has a certain degree of light transmission on the basis of retaining most of the natural color and texture, then in the home industry, transparent wood can also become a highly functional decorative material. This concept increases the application of wood modification in some fields [8].

The process of preparing transparent wood mainly involves the acquisition of delignified wood template and the impregnation of refractive-index-matched resin. It is not difficult to find that delignification is usually an important step in the preparation of transparent wood. Regarding lignin, it accounts for about 30% of the wood mass fraction, and cross-linking with different polysaccharides in the wood increases the mechanical strength of the wood. Components such as lignin and wood extracts cause the wood to undergo strong light scattering and light absorption in the visible light range [9], so that the wood shows color and texture. Because the purpose of this experiment is to prepare transparent wood with color and texture (not absolutely transparent), it is necessary to retain a part of the lignin to achieve the effect of color development, that is, to study the process of removing part of the lignin. In fact, it takes a long time and a lot of chemicals to remove all the lignin [10]. In addition, the removal of all lignin will weaken the wood structure, mainly for the wood template, such as pine, poplar and other low-density tree species are easily broken after delignification, so it is a great challenge for preparing the transparent wood of large-scale or low-density tree species [11]. Therefore, it is of great significance to explore the partial delignification process for the preparation of transparent wood. Although the sample to be prepared is transparent in a broad sense, it has color and pattern, and thus is not absolutely colorless and transparent. Therefore, it was called partially transparent in the title. For ease of interpretation, the following “partially transparent wood” is referred to as “transparent wood (TW)”.

It is known in pre-experiments and references that the amount of lignin removal has a direct impact on the performance of transparent wood, but there is almost no systematic research on the specific impact of removing part of the lignin process on the performance of transparent wood, including the related qualitative and quantitative analysis, and most of the literature on transparent wood removes almost all lignin or chromogenic substances. Therefore, the purpose of this experiment is to study the process technology of partial delignification and the performance of the corresponding transparent wood. The above has certain novelty. It is hoped that the transparency of the transparent wood prepared from different tree species can be improved as much as possible on the basis of retaining the aesthetics, texture and mechanical strength of the wood. In this experiment, the orthogonal test method was used to explore the best technological conditions of removing part of the lignin in the process of preparing transparent wood.

## 2. Experimental Section

### 2.1. Materials

Here, common tree species (one example of dark tree species and one light tree species) are selected: Betula alnoides (*Betula*) and New Zealand pine (*Pinups radiata D. Don*), which are representative tree species in hardwood and coniferous wood respectively. The wood veneers of Betula alnoides and New Zealand pine that are both from Yihua Lifestyle Technology Co., Ltd., China., and these wood veneers are produced in the actual production line. In the experiment, the wood samples were cut to the size of 20 mm (length)×20 mm (width)×0.5 mm (thickness). The physical properties of the two tree species such as air-dry density relative, moisture content and thickness are shown in Table 1. Chemical reagents (analytical grade) used are as follows: ethanol absolute, acetic acid (CH_3_COOH) and sodium hydroxide (NaOH) were all produced by Nanjing Chemical Reagent Co., Ltd. Methyl methacrylate (MMA) and sodium hypochlorite (NaClO_2_) were supplied by Shanghai Macklin Biochemical Co., Ltd. Azobisisobutyronitrile (AIBN) was supplied by Tianjin Benchmark Chemical Reagent Co., Ltd. 

### 2.2. Experimental Methods

In the experiment, acid delignification was applied, and chlorite method was specifically used to prepare the wood template. After obtaining the wood template, a refractive-index-matched resin was impregnated to make the transparent wood. The orthogonal test method is a scientific and effective method for selecting the optimal scheme in multi-factor experiments. With the least number of tests and the most appropriate test method, an optimal test condition and the best scheme will be obtained [12]. Therefore, in the process of removing part of the lignin by the acid method, the orthogonal test method was selected in the experiment to explore the better process conditions for the preparation of transparent wood. With regard to the design of orthogonal experiments, combined with the reading reference of relevant literature and the experience of preliminary experiments, the following three representative factors are selected in the experiment: NaClO_2_ concentration (X), reaction temperature (Y), reaction time (Z). The formulation of specific experimental factor levels is shown in Table 2.

According to the principle of orthogonal experiment design method [13], the orthogonal table L_9_ (3^4^) is used to arrange the three factors/three levels of tests (L_9_ means that nine experiments are needed, at most four factors can be observed, and each factor is three levels). Better production conditions can be determined from these nine sets of data [14]. Table 3 is the proposed orthogonal test table. Therefore, each tree species corresponds to nine different wood templates.

### 2.3. Preparation of Transparent Wood

The following preparation process is shown in Figure 1.

#### 2.3.1. Preparation of Wood Templates

Betula alnoides and New Zealand pine were dried at 103 °C for a few minutes until they were completely dry, and then stored in a drying dish. Subsequently, a certain concentration of NaClO_2_ solution was prepared, and the pH value was adjusted to 4.6 by CH_3_COOH. The samples of the two tree species were heat-treated according to the values in the orthogonal experiment table, and wood templates under different experimental conditions were obtained. The above treated wood templates were washed with deionized water and stored in the ethanol absolute solution. The ethanol absolute solution can displace the residual water in the wood, thus greatly improving the permeability of wood templates.

#### 2.3.2. Preparation of the Polymer

The polymerization inhibitor inside the pure MMA monomer was removed by NaOH solution, and then the MMA was prepolymerized in the water bath at 75 °C, in which 0.38 wt% AIBN was used as the initiator. After 15 minutes, the prepolymerized MMA was cooled to room temperature in the ice water bath to terminate the reaction.

#### 2.3.3. Obtainment of the Transparent Wood

Take out the wood templates stored in the ethanol absolute solution, and let the wood templates and the prepolymerized MMA fully infiltrate for 0.5–1 h under vacuum. Then, the resin-infiltrated wood template was sandwiched in a mold formed by two glass slides and covered them with aluminum foil paper. Finally, the samples were put into an oven at 70 °C for 5 h to complete the further polymerization reaction [15]. The above preparation process was repeated several times to obtain transparent wood under different experimental conditions [16]. In order to facilitate the following experimental test analysis, Betula alnoides is abbreviated as A and New Zealand pine is abbreviated as B; untreated wood samples are collectively referred to as OW, wood templates after removing some lignin are collectively referred to as FW, and the transparent wood after impregnation is collectively referred to as TW. Taking Betula alnoides as an example, untreated wood is OW-A, and the wood templates treated with NaClO_2_ solution (one-to-one corresponding to the setting level of orthogonal test) are labeled FW-A-1, FW-A-2, FW-A-3, FW-A-4, FW-A-5, FW-A-6, FW-A-7, FW-A-8, FW-A-9; the transparent wood made from the corresponding wood templates are named TW-A-1, TW-A-2, TW-A-3, TW-A-4, TW-A-5, TW-A-6, TW-A-7, TW-A-8, TW-A-9. New Zealand pine (B) is marked in the same way as Betula alnoides, except that A is replaced by B.

## 3. Performance Testing

### 3.1. Three Elements Test

The experiment explores the influence of partial delignification process on the properties of transparent wood, and the main components of wood were analyzed qualitatively and quantitatively at each orthogonal test level through the test of three elements (lignin, cellulose, and hemicellulose), in order to refine and compare the corresponding sample characteristics under different process parameters. In the experiment, the contents of three elements in Betula alnoides and New Zealand pine raw samples OW, wood templates samples FW were measured by the National Renewable Energy Laboratory (NREL) method [17].

### 3.2. Color Difference Test

The color reader (PANTONE) was used to measure the color difference between Betula alnoides and New Zealand pine OW, TW samples, and the changes and causes of wood color before and after the experiment were compared and analyzed [18]. The color of each tree species sample is kept as consistent as possible to reduce the test error. Each sample is tested for at least 3-5 replicates and the average value is obtained. The values of the color difference test for the samples are respectively expressed by three parameters L, a and b: L value reflects the brightness, the higher the value is, the higher the brightness is; a value reflects the red-green degree, the positive value is red, the negative value is green; b value reflects the orange-blue degree, the positive value is orange, the negative value is blue; the larger the absolute value of a and b is, the darker the color is [19].

### 3.3. Light Transmittance Test

The light transmittance of two tree species OW and TW were measured by the ultraviolet-visible spectrophotometer (Shanghai youke UV1900PC) at 350–800 nm wavelength. Samples under the same experimental conditions were selected for more than two repeated tests to reduce experimental errors.

### 3.4. SEM Test

The micro morphology of two tree species OW, DW and TW samples were observed by a FEI Quanta 200 scanning electron microscope (SEM). In the experiment, the wood samples were tangential sections obtained by cutting longitudinally along the trunk. In order to observe the change of wood duct or tracheid during the experiment, the samples were cut along the thickness direction by ultra-thin microtome. 

### 3.5. Specific Surface Area and Pore Size Distribution Test

We used an automatic specific surface area and pore size distribution tester (ASAP2020) to test Surface area and Adsorption average pore width of OW (original wood) and FW (wood templates). The relationship between the partial removal of lignin and the distribution of pores in the samples was discussed, as well as the specific effect of changes in pores in these wood templates on light transmittance.

### 3.6. Fourier Infrared Test

The infrared absorption spectrum was obtained by Fourier transform infrared spectroscopy (FTIR), and then the group characteristics and changes of OW, FW and TW were compared and analyzed by the characteristic absorption peaks presented in the figure, so as to carry out the qualitative analysis of related chemical components.

### 3.7. Mechanical Performance Test

The mechanical properties of wood before and after the experiment were tested and calculated. A computer-controlled electronic universal testing machine (SANS-CMT6104) was used to measure the mechanical tensile properties of OW and TW. The upper and lower clamps of the testing machine first fixedly clamped the sample, and set no additional load at this time (to reduce the experimental error), then the lower clamp was fixed, the upper clamp stretched the sample in the direction of the wood grain until it broke, and the upward stretching speed was set to 5 mm/min.

The following formulas are used in mechanical tests:(1)$$\sigma =\frac{F}{S}$$
(2)S=b×h

In the formula, *σ* is the tensile strength; *F* is the maximum force borne by the specimen when it is broken; *S* is the original cross-sectional area in the tensile direction of the sample; *b* is the initial width of the tensile section of the sample, and *h* is the initial thickness of the tensile section [20].

## 4. Results and Discussion

### 4.1. Three Elements Analysis

According to the National Renewable Energy Laboratory (NREL) method and the calculation of fixed formula template (included in NREL), the three major elements content of natural wood samples and delignified wood templates were obtained. The samples tested here need to be processed into 20–80 mesh wood flour and dried until absolutely dry, and it should be noted that the total lignin content here is the sum of acid-insoluble lignin and acid-soluble lignin content [21,22]. The contents of lignin, cellulose, and hemicellulose in Betula alnoides and New Zealand pine OW (original wood) and FW (wood templates) samples are as shown in Figure 2. Generally, wood is mainly composed of cellulose, hemicellulose and lignin, accounting for more than 90% of the total amount of wood, in which cellulose and hemicellulose are colorless substances with simple structure; relatively speaking, the structure of lignin is more complex, which is also one of the main factors of wood coloration [23]. 

The abscissa 1–9 in Figure 2 respectively corresponds to 9 levels of the orthogonal test (0 is the original wood sample), and each level records the specific content of the remaining three elements of the wood sample after partial delignification. It can be seen from Figure 2 that in the process of removing lignin, the contents of the three major elements are all decreased to varying degrees, indicating that the removal of lignin will have a certain impact on the contents of cellulose and hemicellulose. Because the discussion is centered on the removal of part of the lignin, the core of the process exploration is also closely related to the lignin. Therefore, for the two species of Betula alnoides and New Zealand pine, the multi-factor orthogonal range analysis method [24] is combined with the lignin content after the experiment. As a result, Table 4 was obtained. Through the orthogonal range analysis method, the influence of various process parameters on the partial delignification of each tree species in the multi-factor experiment can be obtained, which is meaningful for the further discussion of the process: it can be obtained from Table 4, for Betula alnoides, the influence order of each factor on the lignin content is Y > X > Z, that is, the reaction temperature > NaClO_2_ concentration > reaction time; for New Zealand pine, the order of influence of various factors on lignin content is Z > X > Y, that is, reaction time > NaClO_2_ concentration > reaction temperature.

### 4.2. Color Difference Analysis

For the analysis of color difference values of the above tree species OW, FW and TW, although the specific values are different, the change trend is the same. Therefore, Figure 3 (0 is ow, 1–9 corresponds to nine levels of orthogonal test) is drawn with Betula alnoides as the representative. For the wood template FW, it can be seen from Figure 3 that the remaining lignin content is positively correlated with a and b values, and negatively correlated with L values. It is known that the lignin is one of the main causes of wood color, and the higher the content of lignin, the more orange it is [25]. The positive value of b represents orange, and the larger the value is, the darker the color is. Therefore, the change trend of lignin content and b value is consistent with the theory, that is, the more the residual lignin content of FW is, the greater the b value is. Because the reagent used to remove partial lignin is sodium hypochlorite, which has certain bleachability, the whiteness of the wood template increases, the brightness also increases, and the L value increases. For the TW of the two tree species, the filling of the transparent resin improves the lightness of the wood, and the L value is higher than that of the original wood OW, and lower than that of FW. The a values are slightly reduced, and the b values are increased compared to FW. It can be seen that the filling of the resin not only makes the wood have a certain degree of light transmission, but also retains most of the color and texture of the wood.

Figure 4 shows the samples of level 5 and 9 in the orthogonal test, because the amount of delignification in level 5 tends to average compared with the whole, and the content of delignification in level 9 is the highest, which is very representative. The sample with certain light transmittance is placed on the paper printed with "NFU" under the sunlight; and under the condition of specific light source, the light transmittance shows the unique texture and color of wood, which is very novel and beautiful, showing the feasibility of the process and the potential of the material as a functional decoration material.

### 4.3. Light Transmittance Analysis

Figure 5a,b show the light transmittance of Betula alnoides and New Zealand pine OW and TW samples respectively. The numbers 1–9 correspond to the 9 levels of orthogonal test (0 is the original wood sample). The light transmittance of the original wood of the two tree species is very low in the visible light wavelength range, which is mainly due to the light absorption of the color-producing components such as lignin, and also includes the light scattering caused by the porous structure of wood [26]. It can be seen from Figure 5 that the OW light transmittance of the darker Betula alnoides with a higher density is slightly lower than that of the lighter New Zealand pine with a lower density. After the experiment, the light transmittance of TW of the two tree species has been improved compared to the OW, and the light transmittance of New Zealand pine has increased more than Betula alnoides: For Betula alnoides, TW-A-9 has a maximum increase of 11.39% at 800 nm; for New Zealand pine, TW-B-9 has a maximum increase of 14.14% at 800 nm. At the same time, it can be found that the light transmittance values corresponding to 1–9 are negatively correlated with the remaining lignin content, that is, the lower the remaining lignin content is, the more the light transmittance increases. The reason for this is that the more lignin is removed from the wood template, the more pores there are, and the more transparent resin is infiltrated, so the higher the light transmittance of the sample is [27]. In this test, the performance is explored by the process, and the process is deduced from the performance. The goal is to hope that the light transmittance of the wood made of different tree species can be improved as much as possible while retaining the beauty, texture, and strength of the solid wood. Here, it can be concluded that the optimal experimental level for Betula alnoides is 9; the optimal experimental level for New Zealand pine is 9.

### 4.4. SEM Analysis

Taking orthogonal test levels 5 and 9 as examples for analysis, since the amount of lignin removed by level 5 tends to be average compared with the whole, and the content of lignin removed by level 9 is the highest, which are all very representative, the observed sample is OW-A, FW-A-5, FW-A-9, TW-A-5, TW-A-9; OW-B, FW-B-5, FW-B-9, TW-B-5, TW-B-9. From the analysis of the three major elements (lignin, cellulose, and hemicellulose), it can be seen that most of the lignin is still retained, so the difference between the wood templates FW of the same tree species is not large. Figure 6a,f present the micro-morphology of OW-A and OW-B respectively, showing the natural porous structure of the wood. It is known that lignin is mainly concentrated in the cell corner intercellular layer, as is the case for birch and New Zealand pine. In the process of removing part lignin, cracks appear in the triangle area where three cells intersect due to the decrease of lignin [28], as shown in Figure 6b,c,g,h. From FW to TW of two tree species, after impregnation polymerization, it was observed that PMMA not only fully filled the cracks but also penetrated into the cells of wood. The honeycomb porous structure of the wood was almost eliminated, and the cell walls were more closely bonded, which indicated that the wood template and PMMA had a good synergistic effect, and played a role in replacing part of the lignin to connect the cellulose skeleton to a certain extent.

### 4.5. Specific Surface Area and Pore Size Distribution Analysis

The BET specific surface area refers to the total area of a unit mass of material [29]. It can be obtained from Figure 7a,b that the BET specific surface area value and the average adsorption pore size value have a negative correlation as a whole. At the same time, the more the delignification content is, the larger the specific surface area is, and the smaller the average adsorption pore diameter is. This is because the original adsorption pore diameter is mostly from the larger conduit hole in the wood. When part of the lignin is removed, there will be a lot of nano-sized micro pore diameters in the wood template, so the average adsorption pore size value decreases, and the specific surface area increases. It is not difficult to find that both the Betula alnoides and New Zealand pine have the highest specific surface area value at experimental level 9. This is because the wood template (level 9) has more pores. Owing to the increase of wood template pores, the value of specific surface area increases and PMMA has better penetration effect. Therefore, for both tree species, orthogonal test levels 9 has the highest light transmittance, corresponding to the above.

### 4.6. Fourier Infrared Analysis

Figure 8a,b shows the infrared spectrum of OW and FW for two tree species. The characteristic absorption peaks of OW-A and OW-B include 3417 cm^−1^ (O-H stretching vibration in cellulose), 2920 cm^−1^ (C–H stretching vibration), 1735 cm^−1^ (Acetyl site of hemicellulose), and 1504 cm^−1^ (stretching vibration of aromatic skeleton group of lignin), 1165 cm^−1^ (C–O–C stretching vibration in cellulose) [30]. Although Betula alnoides and New Zealand pine are two different tree species, their main components are similar as wood, so the main stretching vibrations of the spectrum are basically the same. FW1-9 (including two tree species) correspond to the 9 levels of the orthogonal test respectively. It is found that the peak intensity of FW weakens at 3417 cm^−1^, 1735 cm^−1^, and 1504 cm^−1^, which proves that the content of lignin in the samples was indeed decreasing [31]. Partial lignin has been removed, and the content of cellulose and hemicellulose has also been slightly reduced, which is in line with the test results of the three major elements. Figure 8c,d are the infrared spectra of TW from Betula alnoides and New Zealand pine. After resin impregnation polymerization, TW not only has part of the characteristic absorption peak of the wood, but also has the characteristic peak of PMMA (2992^−1^ and 2950 cm^−1^ for C−H, 1740 cm^−1^ for C=O, and 1191 and 1145 cm^−1^ for C−O) [32], which shows that PMMA and wood template have good synergistic effect. 

### 4.7. Mechanical Performance Analysis

Under the load of the longitudinal tension, the samples went through the elastic deformation stage, the yield stage, the plastic deformation stage and instantaneous fracture stage. With the moment of fracture, the load value dropped sharply. Therefore, the force that the sample bore at the moment of fracture was the load value F corresponding to the highest point of the curve. Both OW and TW (1–9 corresponding to 9 levels of orthogonal test) of Betula alnoides and Pinus New Zealand were tested to compare the mechanical properties of the original wood and the impregnated wood samples. After each sample was tested, the fracture load F (taking the average of multiple tests) was obtained, and then the tensile strength σ value was calculated according to the formula, as shown in Figure 9, where the σ value represents the maximum bearing capacity of the sample under static tensile condition. It can be seen from Figure 9 that the tensile strength and ductility of all TW of Betula alnoides and New Zealand pine are higher than that of OW: the maximum increase rate is 57.58% for Betula alnoides and 57.76% for New Zealand pine. This shows that PMMA has a good synergy with the partially delignified wood templates from a macro perspective, so the overall mechanical properties have been improved.

## 5. Conclusions

Through the study on the performance of the transparent wood under the process of removing part of the lignin, it is concluded that this kind of transparent wood has a certain degree of light transmittance while retaining most of the wood color and texture. The light transmittance of some tree species is increased by 14.14%, and the mechanical strength is increased by 57.76%, demonstrating that it has good development prospects in the home furnishing industry and will become a highly functional decorative material. At the same time, it can be concluded that the amount of lignin removal has a direct impact on the performance of transparent wood. In the test, combined with the orthogonal test method, the qualitative and quantitative analysis is carried out, and a good process plan is selected. In addition, although the specific values of the two selected tree species are different in the test, the overall trend of change is basically the same, which shows that this experiment is applicable to a variety of wood and is practical, and worthy of detailed further discussion.

## Figures and Tables

**Figure 1 polymers-12-00661-f001:**
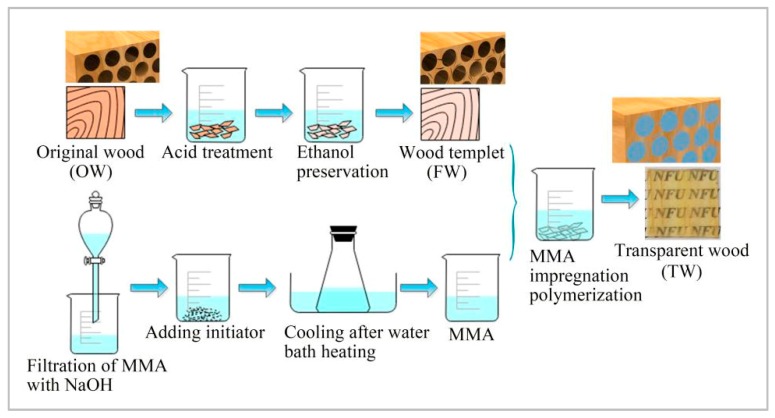
The main preparation process of transparent wood (TW).

**Figure 2 polymers-12-00661-f002:**
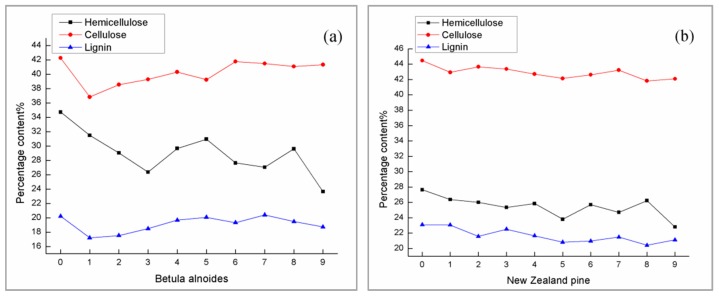
The contents of lignin, cellulose, and hemicellulose in (**a**) Betula alnoides and (**b**) New Zealand pine OW and FW samples.

**Figure 3 polymers-12-00661-f003:**
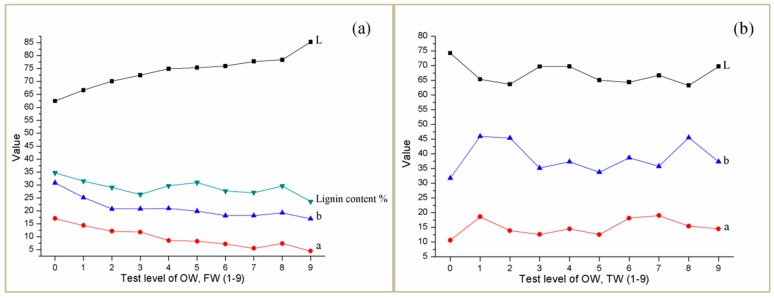
The color difference of OW, (**a**) FW and (**b**) TW.

**Figure 4 polymers-12-00661-f004:**
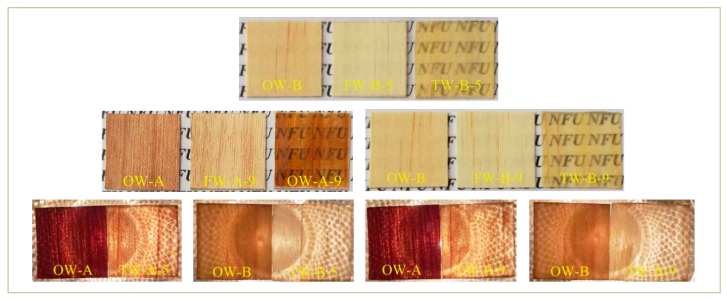
Contrast photos of OW-A, FW-A-5, TW-A-5, OW-B, FW-B-5, TW-B-5, OW-A, FW-A-9, TW-A-9; OW-B, FW-B-9, TW-B-9.

**Figure 5 polymers-12-00661-f005:**
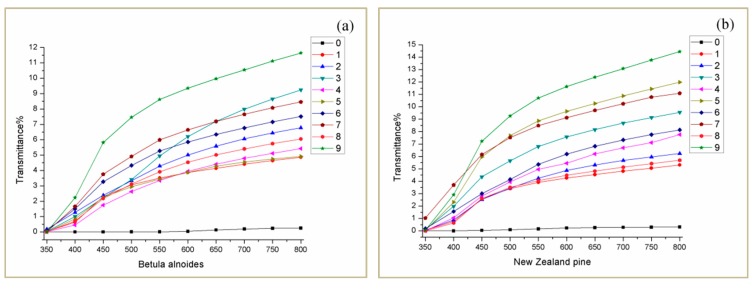
The light transmittance of OW and TW for two tree species (**a**,**b**).

**Figure 6 polymers-12-00661-f006:**
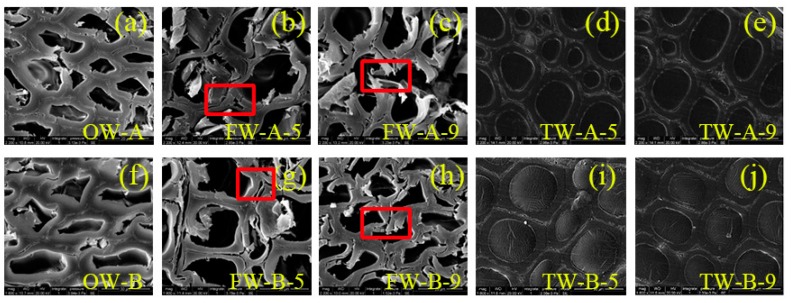
The micro morphology of (**a**) OW-A, (**b**) FW-A-5, (**c**) FW-A-9, (**d**) TW-A-5, (**e**) TW-A-9; (**f**) OW-B, (**g**) FW-B-5, (**h**) FW-B-9, (**i**) TW-B-5, (**j**) TW-B-9.

**Figure 7 polymers-12-00661-f007:**
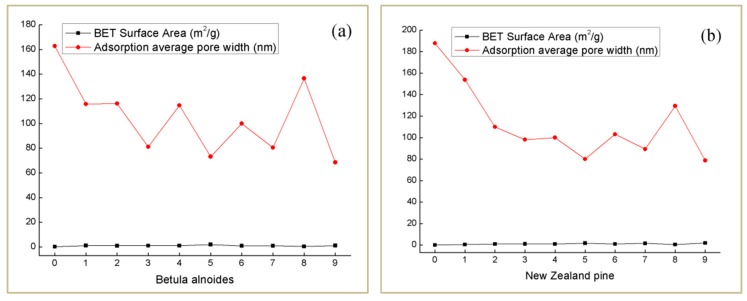
Surface area and adsorption average pore width of OW and FW for two tree species (**a**,**b**).

**Figure 8 polymers-12-00661-f008:**
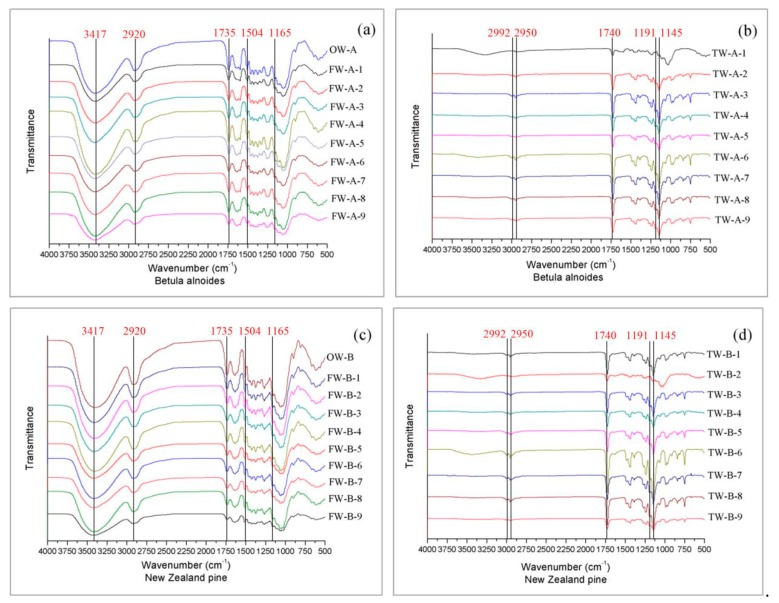
The infrared spectrum of OW, FW and TW for two tree species corresponding to (**a**–**d**).

**Figure 9 polymers-12-00661-f009:**
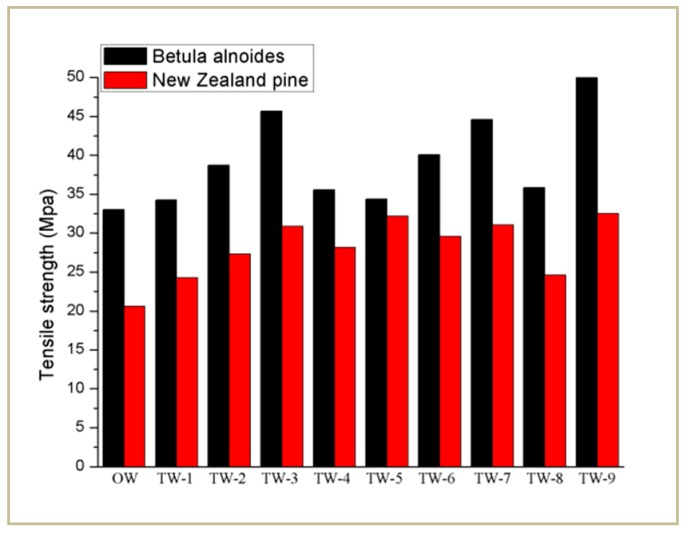
The tensile strength of OW and TW for two tree species.

**Table 1 polymers-12-00661-t001:** Physical properties of two tree species.

Wood Species	Air-Dry Density Relative (g/cm^3^)	Moisture Content (%)	Thickness (mm)
Betula alnoides (A)	0.65	9.96	0.50
New Zealand pine (B)	0.31	10.01	0.50

**Table 2 polymers-12-00661-t002:** Factor level of the orthogonal experiment.

	X/wt%	Y/°C	Z/min
1	0.4	70	45
2	0.7	80	90
3	1	90	135

**Table 3 polymers-12-00661-t003:** Orthogonal test table on the preparation of wood templates.

	XYZ	X (wt%)	Y (°C)	Z (min)
1	111	0.4	70	45
2	122	0.4	80	90
3	133	0.4	90	135
4	212	0.7	70	90
5	223	0.7	80	135
6	231	0.7	90	45
7	313	1	70	135
8	321	1	80	45
9	332	1	90	90

**Table 4 polymers-12-00661-t004:** Orthogonal test results and range analysis.

Betula Alnoides			
Lignin content/%	X	Y	Z
Mean value1	28.95	29.39	29.57
Mean value2	29.42	29.86	27.45
Mean value3	26.77	25.88	28.12
Range	2.65	3.98	2.12
**New Zealand Pine**			
Lignin content/%	X	Y	Z
Mean value1	25.91	25.64	26.10
Mean value2	25.11	25.34	24.88
Mean value3	24.58	24.61	24.61
Range	1.33	1.03	1.49

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
