# Peer review of "Study on the Properties of Partially Transparent Wood under Different Delignification Processes"

_polymers, 2020, doi:10.3390/polym12030661_

Round 1
Reviewer 1 Report
Dear authors,
thank you for your revisions based on my comments in the previous version of your manuscript. I suggest accepting your manuscript.
Kind regards
Lubos Kristak
Author Response
Dear Editors,
Thanks again to the reviewer's valuable comments on “Study on the Properties of Partial Transparent Wood under Different Delignification Process”, it is our honor. We have sent the revised manuscript and have made changes by using the “track change mode”. The following are the point-by-point response and the corresponding revisions.
Reviewer: 1 Recommendation: Accept.
Comments: Thank you for your revisions based on my comments in the previous version of your manuscript. I suggest accepting your manuscript.
Answer: We sincerely thank you for your comments and suggestions for improving the quality of our article. Thank you again for your valuable suggestions and it is our honor. We attach great importance to it and make corresponding modifications.
Thank you and best regards.
Yours sincerely,
Yan Wu and Jichun Zhou

Reviewer 2 Report
This paper reports the Study on the Properties of Partial Transparent Wood under Different Delignification Process.
The investigation is really interesting and impressive. Moreover, this manuscript is well organized with experimental data and analysis methods. Therefore, I have no major problems with this work. Accordingly, I recommend this manuscript for publication in polymers after only minor revision. The following issues should:
manscript is too long
Please describe the scientific novelty carefully
Please, better describe the methodology of mechanical properties, FTIR Figure 5 and 8 is illegible,please describe them better
Author Response
Dear Editors:
Thanks again to the reviewer's valuable comments on “Study on the Properties of Partial Transparent Wood under Different Delignification Process”, it is our honor. We have sent the revised manuscript and have made changes by using the “track change mode”. The following are the point-by-point response and the corresponding revisions.
Reviewer: 2
Recommendation: Minor revisions.
Comments: The investigation is really interesting and impressive. Moreover, this manuscript is well organized with experimental data and analysis methods. Therefore, I have no major problems with this work. Accordingly, I recommend this manuscript for publication in polymers after only minor revision. The following issues should:
- Manuscript is too long.
Answer: Yes, it has been improved. Some redundant descriptions have been deleted. In the article, the orthogonal test method was used to explore the better process conditions for the preparation of transparent wood. The tests of color difference, light transmittance, porosity, microstructure, chemical groups, mechanical strength were carried out on the wood templates and transparent wood under different experimental conditions. In addition, through the three major elements (lignin, cellulose, hemicellulose) test and orthogonal range analysis method, the influence of each process factor on the lignin removal of each tree species was obtained. The comprehensive analysis of the two tree species above also makes the article longer.
- Please describe the scientific novelty carefully.
Answer: Yes, it has been improved. Indeed, it is necessary to add descriptions of the scientific novelty. Explained and carefully emphasized on the cutting edge: It is known from the pre-experiments and references that the amount of delignification has been a direct impact on the performance of transparent wood, but there is almost no systematic research on the specific impact of partial removing lignin process on the performance of transparent wood, including the related qualitative and quantitative analysis, and most of the literature on the transparent wood removed almost all the lignin and chromogenic substances. Therefore, the purpose of our experiment is to study the process technology of partial delignification and the performance of the corresponding transparent wood. The above has certain novelty.
- Please, better describe the methodology of mechanical properties, FTIR Figure 5 and 8 is illegible,please describe them better.
Answer: Yes, it has been modified. Your suggestions are valuable. It is true that the previous description of “the methodology of mechanical properties” is not good and has been changed to “The upper and lower clamps of the testing machine first fixedly clamped the sample, and set no additional load at this time (to reduce the experimental error), then the lower clamp was fixed, the upper clamp stretched the sample in the direction of the wood grain until it broke, and the upward stretching speed was set to 5mm/ min”. Indeed, it is necessary to describe Figures better. According to the principle of orthogonal experiment design method, the orthogonal table L9 (34) is used to arrange the three factors / three levels of tests. (L9 means that nine experiments are needed, at most four factors can be observed, and each factor is three levels.). Then the better production conditions can be determined from these nine sets of data. Therefore, each tree species corresponds to nine different wood templates. Therefore, each tree species in the picture corresponds to a series of different samples. In fact, the multiple curves in the figures mainly reflect the trend and contrast issues, and are not limited to each specific value. Sometimes it is more inclined to express a trend. For example, Figure 5 is about light transmittance and mainly uses to express that the values of curves TW1-9 are much higher than that of curve 0 (OW), and curves TW1-9 have some values that overlap, which is unavoidable and not the focus of discussion. Of course, for clearer expression of some curves, we have replaced the line format and expression method, and redrawn to improve the picture resolution for better description (please see the new Figure 5). In particular, for the infrared image, the text makes a detailed explanation of the highlighted line with labeled values in the new Figure 8 to achieve the perfect combination of images and text.
Thank you again for your valuable suggestions and it is our honor. We attach great importance to it and make corresponding modifications.
Thank you and best regards.
Yours sincerely,
Yan Wu and Jichun Zhou

Round 2
Reviewer 2 Report
I accept the manuscript